# Karyotype Variability in Wild *Narcissus poeticus* L. Populations from Different Environmental Conditions in the Dinaric Alps [note 1]

**DOI:** 10.3390/plants13020208

**Published:** 2024-01-11

**Authors:** Fatima Pustahija, Neđad Bašić, Sonja Siljak-Yakovlev

**Affiliations:** 1Faculty of Forestry, University of Sarajevo, Zagrebačka 20, 71000 Sarajevo, Bosnia and Herzegovina; 2Ecologie, Systématique et Evolution, Université Paris-Sud, CNRS, AgroParisTech, Université Paris-Saclay, bâtiment 680–12, route 128, 91190 Gif-sur-Yvette, France

**Keywords:** B chromosomes polymorphism, chromosome rearrangement, genome size, karyotype, rDNA and heterochromatin organization

## Abstract

*Narcissus poeticus* L. (Amaryllidaceae), a facultative serpentinophyte, is a highly variable species and particularly important ancestor of cultivated daffodils, but is rarely studied in field populations. This study, based on natural populations in the Balkans, focused on karyotype variability, genome size, ploidy and the presence of B chromosomes. Thirteen native populations from different environmental and soil conditions were collected and analyzed using flow cytometry to estimate nuclear genome size, fluorescence in situ hybridization (FISH) for physical mapping of rDNA, fluorochrome labeling (chromomycin and Hoechst) for heterochromatin organization and silver nitrate staining of nucleoli for determining rRNA gene activity. The organization of rDNA and natural triploids is reported here for the first time. The presence of individuals with B chromosomes (in 9/13 populations) and chromosomal rearrangements was also detected. The observed B chromosome showed three different morphotypes. The most frequent submetacentric type showed four different patterns, mainly with active ribosomal genes. The results obtained show that *N. poeticus* has a dynamic genome with variable genome size due to the presence of polyploidy, B chromosomes and chromosomal rearrangements. It is hypothesized that the observed changes reflect the response of the genome to different environmental conditions, where individuals with B chromosomes appear to have certain adaptive advantages.

## 1. Introduction

The genus *Narcissus* (Amaryllidaceae) is native to Central Europe and the Mediterranean region of Europe and North Africa, including one species in Asia, with a center of diversity in the Iberian Peninsula [1]. The taxonomy of the genus is very complex, and diverse classifications have been proposed comprising from 22 [2], 27 [3], 36 [4,5] to 81 species [6]. More than 22,000 cultivated varieties have been described [7], mainly derived from *Narcissus poeticus* L., *N. pseudonarcissus* L. and *N. tazetta* L. [8,9]. This genus shows variability in genome size (from 2C = 13.10 to 67.70 pg), in basic chromosome numbers (x = 5 and 7) with several derived numbers and in ploidy levels (2, 3, 4 and 6x). The presence of individuals with aneuploidy and supernumerary chromosomes (B chromosomes, Bs) in a natural environment [5,10,11,12] has also been reported. Based on morphological and cytogenetical investigations [3,5,13,14], the complex of *N. poeticus* (syn. *N. radiiflorus* (Salisb.) Baker, *N. angustifolius* Curtix ex Haw.) can be considered as a species native to Southwest and Southeast Europe.

According to Webb [3], daffodils grow naturally in grasslands and open woods and on river banks and rocky hillsides, whereas for the poet’s daffodil, only mountain meadows have been indicated. Nevertheless, in the karst fields of Bosnia and Herzegovina, the poet’s daffodil is present in very large populations with numerous individuals. According to our observations, this species is also present in smaller populations with a low number of individuals scattered throughout rocky hillsides, mountain and subalpine grasslands developed on limestone and on serpentine substratum (as facultative serpentinophyte), at elevations from 700 to 1570 m. This species requires basic to slightly acidic and moist to well-drained soils, grows in full sun to partial shade and flowers from the end of April to June, with a characteristic and intense fragrance. For all these reasons, *N. poeticus* could be an appropriate natural model for studying genome organization under a variety of environmental and soil conditions.

Despite its importance in the development of the garden-variety narcissi, relatively few studies have been carried out on natural populations. However, various evolutionary processes, such as hybridization, polyploidization, adaptive radiation and presence of Bs have been noticed in the genus *Narcissus* [5,8,13,15,16].

The most frequent level of polyploidy for cultivars of *Narcissus* is tetraploid, while individuals with a higher ploidy level are rare [8,15,16]. Polyploidy is rare in wild populations, with the exception of *N. bulbocodium*, which can present 2x, 4x and 6x [8]. To date, individuals from natural populations of *N. poeticus* have been characterized as diploids, 2n = 14 [5,8,15,16,17]. Triploids (2n = 21) have only been observed in two populations, one from Greece and one from Switzerland [5,8]. Even most of the cultivated varieties of poet’s daffodil are diploids, while some are triploids and one is tetraploid [5,8,15,16,17].

Although they are not necessary for the survival of individuals, B chromosomes are present in 10–15% of angiosperms, mostly in species with large genomes and a small number of chromosomes [18,19,20]. B chromosomes do not pair with A chromosomes, have irregular modes of inheritance and do not follow the principles of Mendelian segregation; they vary among taxa in number, size and composition.

The presence and number of Bs can affect the fitness of the plant, a smaller number of Bs having less effect on the phenotype [18,20,21,22,23,24,25,26,27,28]. In general, a negative effect of a high number of Bs is observed in A chromosomes’ behavior, organism fertility and reduction in plant height and weight, and seed weight [18,22,28]. Nevertheless, a positive effect of Bs on *Allium schoenoprasum* has been reported [29,30], as its seeds carrying Bs germinated better and showed a better survival ability than 0B seeds under drought stress conditions.

In addition, Karafiátová et al. [31] recalculated previously published data for *Secale cereale* and *Zea mays* and concluded that the deleterious effects of B chromosomes on plant fitness are observed when the DNA mass of Bs exceeds 20% of the total nuclear DNA mass, regardless of the number of chromosomes.

The presence of B chromosomes in the genus *Narcissus* has been reported for several species [5,8,15,16,17]. All B chromosomes found were about two-thirds the size of the satellite chromosome, with a subterminal centromere position; were either very similar or morphologically variable; and may or may not be heterochromatic [5,8,15,16]. However, there are only a few references in the literature about the presence of B chromosomes in *N. poeticus* [17].

The amount of nuclear DNA in a species is generally constant, but intraspecific variation has also been recorded. In general, the amount of DNA always increases due to ploidy and the presence of Bs [18,19,20,25,32]. Moreover, numerous studies have shown different relationships between genome size and certain cellular and physiological characteristics [33,34,35,36].

The previously reported genome size for *N. poeticus* ranges from 24.33 pg [37] to 27.50 pg [38] for diploid individuals, from 34.55 pg [39] to 38.70 pg [5] for triploids and is 52.70 pg for the tetraploid cultivar [5].

The aims of this study were as follows: (1) to compare the karyotype features from diverse natural populations of *Narcissus poeticus* in the Western Balkan Peninsula by studying the heterochromatin and rDNA organization; (2) to estimate possible genome size variations among populations from different habitats comprising heavy metal-rich soils (serpentine); and (3) to identify the possible genome response of *N. poeticus* to different environmental conditions.

## 2. Results

### 2.1. General Karyotype Characteristics

In all the populations studied, *N. poeticus* had the diploid chromosome number 2n = 2x = 14, except in two populations (8 and 12; Table 1), where there were also some triploid individuals, 2n = 3x = 21. Inter-individual variations in karyotype were observed in ploidy levels and in the presence of B chromosomes with variable numbers and morphologies.

Morphometric karyotype data are presented in Appendix A. The common chromosome formula of *N. poeticus* is 2n = 8 sm + 2 m + 2 st + 2 m^sat^ (Appendix A; Figure 1E). When observed, B chromosomes were mainly of the submetacentric type, but they can also be metacentric or subtelocentric. The average chromosome length varied from 5.02 to 9.31 µm.

### 2.2. Heterochromatin Organization and Physical Mapping of rRNA Genes

Representative metaphase plates and idiograms showing the positions of 5S and 35S rDNA loci, CMA, Hoechst and DAPI bands of the standard karyotype are shown in Figure 1.

In the standard karyotype of *N. poeticus*, 22–24 CMA signals, depending on whether the satellite in pair 7 was stained or not, were distributed over all chromosomes. Chromosome pairs 1 and 5 shoved only a thin terminal CMA band on the long arm, while on pair 6, a band was located on the short arm. Chromosome pair 3 had only a centromeric CMA band, and chromosome pair 4 possessed weak terminal CMA bands on both arms. An intense intercalary band on the long arm and a weak terminal band on the short arm were observed on chromosome pair 2. The satellite chromosome pair 7 was characterized by a centromeric band and one weak terminal band on the long arm, not always visible. In addition, this chromosome pair displayed very intense CMA bands on both sides of the secondary constriction (SC), including the satellite, or only on the terminal part of the short arm, without satellite staining (Figure 1E, boxed).

Hoechst staining showed that only chromosome pair 2 had AT-rich DNA regions: one weak centromeric band and more intense intercalary bands on the long arm, which colocalized with the CMA band (Figure 1(A2), blue arrows).

The intercalary DAPI bands, after the FISH experiment, were visible only on chromosome pair 2 (long arm, corresponding to heterochromatin region) and on the short arm of chromosome pair 4, colocalized with 5S rDNA loci (Figure 1(A2,E)).

All triploid individuals displayed the same standard karyotype as diploids.

Fluorescent in situ hybridization results for *N. poeticus* were reported for the first time in this study. The standard karyotype (Figure 1(A2–D2,E)) showed a 5S rDNA locus on chromosome pair 4 that colocalized with DAPI bands. The 35S rDNA locus was visible as a very intense signal at the terminal part of the short arm or on both sides of the secondary constriction of chromosome pair 7 (Figure 1(A2,B2), red arrows).

Positive DAPI bands after FISH staining appeared slightly brighter than the navy blue color of chromosomes. Their position corresponded to the Hoechst and CMA bands on the long arm of chromosome pair 2 and to the 5S locus signal on the short arm of chromosome pair 4. In both cases, when the images were superimposed, the DAPI signal was masked (Figure 1(A2–D2,E)).

The standard rDNA pattern for diploids is also present in the karyotype of triploid individuals (Figure 1(D1,D2,E)).

### 2.3. Particular B Chromosome Types in N. poeticus

One to three B chromosomes were observed in the majority of the populations investigated (in 9 out of 13 populations). The average DNA content of the B chromosome in diploid individuals with one Bs was 2.22% and with two Bs was 5.03%, while in triploid individuals with three B chromosomes, Bs represented 12.30% of the total genome size (Table 2; (%Bs = (2C DNA value with Bs − 2C DNA value without Bs) ∗ 100/2C DNA value without Bs)).

These accessory chromosomes were characterized by different patterns of GC-rich regions and by the presence of 35S rRNA genes. The most frequent submetacentric type of B chromosomes in *N. poeticus* (type I) showed four different patterns (Figure 2). Only four individuals analyzed had B chromosomes without signals. All other submetacentric B chromosomes showed CMA signals collocating with 35S rRNA genes on the long arm. Three combinations of signals corresponding to this colocalization could be seen: one intense band (type Ib), a large area where the CMA and 35 rDNA signals are dispersed (type Ic), or a combination of these two patterns (type Id) (Figure 2). The metacentric B chromosome (type II) had CMA bands colocalized with 35S rRNA genes on the long arm, in the form of an intense subtelocentric band, and on the short arm in the form of dispersed subtelocentric signals (Figure 2). The subtelocentric B chromosome (type III) is characterized by dispersed CMA bands, always colocalized with 35S rRNA genes, on a proximal position of the long arm near the centromere (Figure 2).

The triploid individual from population 8 possessed three morphologically different types of B chromosomes: type I, II and III (Figure 2; Table 1). Four diploid individuals had Bs without constitutive heterochromatin and rRNA genes (type Ia), while one individual had Ib, eight individuals Ic and three individuals Id types of B chromosomes (Figure 2; Table 1). Individuals from serpentine populations (11, 12 and 13) were characterized by the presence of 1–2 submetacentric B chromosomes of types Ib, Ic and Id.

It is interesting to note that the 33 individuals analyzed from the submountain and subalpine meadows did not have any B chromosomes and that seven of these individuals showed colocalization of the CMA band and the 35S rDNA locus just on the terminal part of the short arm of chromosome pair 7 (Table 2, Figure 1(A1,A2,B1,B2,E), boxed).

The activity of rRNA genes located on B chromosomes was confirmed by the number of nucleoli observed after silver nitrate staining and by observation of these genes in interphase nuclei after FISH (Appendix A). The number of nuclei formed increased and corresponded to the total number of 35S rDNA loci found on A and B chromosomes in the case of individuals in which the Bs carried rRNA genes.

All the cytogenetic characteristics of the individuals analyzed were stable and were also observed after three or four years of culture in the greenhouse at the University of Paris-Sud, France.

### 2.4. Deviations from the Standard Karyotype Features

In three populations, some individuals with different types of chromosomal translocations or inversions were observed: two on serpentine and one on limestone (Appendix A).

One type of deviant karyotype was observed in an individual bearing B chromosomes from population 11 on a serpentine substrate (Appendix A). This karyotype is characterized by the inversion of a large part of the chromosome on one homolog of pair 4. This karyotype showed the same CMA and Hoechst patterns as the individuals with the standard karyotype.

Two types of chromosomal rearrangement observed in populations 2 (subalpine limestone meadow) and 12 (serpentine) showed clearly different karyotypes in relation to the common feature. For the serpentine individual (Appendix A), it was possible to complete only six chromosome pairs, with two remaining metacentric chromosomes characterized by important differences in length. In the second case of translocation, only five pairs of chromosomes could be completed (Appendix A). Individuals with translocations have almost the same Hoechst, 5S and 35S positions as in the standard karyotype. Only the individual from the serpentine population was characterized by one additional terminal unpaired 35S rDNA locus on the long arm of chromosome pair 3 (Appendix A).

### 2.5. Genome Size Estimation

The DNA content of 13 populations was determined and presented as mean values in Table 2. Measurements were performed in diploid plants without Bs and with 1–2 Bs, in triploid plants without B and with three B chromosomes and in the plants with chromosomal translocations. Cytometric measurements carried out on leaves, bulbs and roots did not reveal any major variations in genome size within populations. The only inter-population variations were linked to the presence or absence of B chromosomes and triploidy. The peaks observed in histograms always corresponded to 2C nuclei, without endopoliploidy or even 4C nuclei (Appendix A).

Among the populations studied, the genome size for diploids without B chromosomes ranged from 23.97 to 25.84 pg, for diploids with Bs from 24.31 to 26.86 pg, for triploids without Bs from 34.51 to 35.32 pg and for the triploid individual with 3Bs the genome size was 38.80 pg. Analysis of variance (ANOVA) showed that there were significant differences between groups. Duncan’s test showed that the inter-population differences of genome size between populations with or without B chromosomes were significant (*p* < 0.05). Intra-population differences in 2C DNA values between diploid individuals with and without Bs were also significant (*p* < 0.01).

## 3. Discussion

### 3.1. Karyotype Characteristics and Heterochromatin Organization

Previous studies have reported that *N. poeticus* and *N. radiiflorus* represent a single species with some morphological differences [3,14,39]. Our results also confirm that *N. poeticus* represents a polymorphic species in terms of variability in karyotype structures and banding patterns.

The standard karyotype of *Narcissus poeticus* described in this study corresponds to those described by Wylie [8], Tucci et al. [14], Brandham and Kirton [16], D’Amato [39], Maugini [40] and De Dominicis et al. [41].

D’Amato [39] and De Dominicis et al. [41] reported the presence of three regions of constitutive heterochromatin after Giemsa staining. Our results confirm that the intercalary band on chromosome pair 2 corresponds to colocalization of GC and AT rich DNA regions and constitutive heterochromatin (DAPI after FISH). The same authors also reported that the intercalary C-band on chromosome pair 4 is not present in all metaphase plates and was not visualized by fluorochrome staining. However, the double FISH experiment showed colocalization of nonspecific heterochromatin (DAPI bands) and the 5S rDNA locus in the same region in our study. The satellite pair, characterized by the presence of a third C-band at the SC level, corresponds to the CMA band and the 35S rDNA locus in our study. According to previous reports [14,39,41], silver staining confirmed the presence of NOR at the SC on this chromosome pair.

All these results refer to the constancy of standard karyotype traits and heterochromatin organization at the specific level despite the geographical distances between populations and the different habitat types in which they grow.

### 3.2. The B Chromosomes in Narcissus poeticus

B chromosomes are a dynamic system, and their frequency depends on two opposing forces: accumulation and its effects on the fitness of individuals [20,21,22,23,24,25,26,27,28]. Individuals in natural populations had no more than three to four B chromosomes, with some exceptions, e.g., *Allium schoenoprasum*, *Pennisetum violaceum*, *Secale cereale*, *Sorghum halepense*, *Sorghum purpureosericeum* [22,23,24,31,42]. In the genus *Narcissus*, especially in the garden varieties, the number of B chromosomes usually varies from one to a maximum of five [8,16], Takhtajan [17] reporting the presence of 1–4 Bs in *N. angustifolius* (syn. *N. poeticus*). In this study, the most frequent number of Bs in diploid individuals was one, less frequently two. In population 8, however, three Bs were found in diploid and triploid individuals.

In our study, the observed number of Bs and their contribution to the total DNA content of poet’s daffodils (Table 1 and Table 2) indicate their weak influence on the fitness of individuals and populations as a whole, but also the possible response to water stress and large temperature fluctuations in the environment in which they live. Although Porter and Rayburn [43] found that the number of Bs in *Zea mays* was not significantly correlated with altitude, the difference in the absence/presence of Bs in poet’s daffodil populations on the continental and in the littoral Dinaric Alps at altitudes above 1000 m above sea level (Table 2 and Table 3) may indicate that there is a correlation between the presence of Bs and different water and temperature regimes.

### 3.3. rDNA and Heterochromatin Organization on B Chromosomes

The B chromosomes observed in the genus *Narcissus* are mostly euchromatic and, with few exceptions, identically stained as the A chromosomes [8,16]. All observed types of B chromosomes were similar in size, about two-thirds as large as the smallest chromosome pair, which is consistent with data reported for another *Narcissus* species with B chromosomes [8].

In this study, the presence of morphological and structural Bs polymorphism is observed in the poet’s daffodil. According to Jones [18], there are two or more types of Bs in more than 65 plant species, the origins of which can be explained by centromere misdivision during meiosis. Therefore, B chromosome types II and III may be ephemeral products of different mutational events on type I (Figure 2). Indeed, the common Bs type (submetacentric) is well conserved and widespread in all populations with Bs studied. Based on the above and data from the literature [44,45,46], we can assume that Bs, regardless of their type, have a similar origin and that they most likely arose through rearrangements of the base chromosome set.

According to Camacho et al. [23], Bs polymorphism is a dynamic system, while Puertas et al. [21] assumed that the Bs themselves are mainly responsible for the maintenance of B chromosome polymorphisms in *Secale cereale* and *S. vavilovii*, and not their co-adaptation with the nuclear genome. In *Allium schoenoprasum* [30,47], the high degree of structural polymorphism is found in the population with the highest frequency of B chromosomes. These authors reported that humidity, particularly drought, had a significant effect on increasing the proportion of non-standard B chromosome types. The greatest polymorphism of poet’s daffodil Bs was found in all populations on serpentine substrates and in two populations on karst fields, that live in very dry conditions in summer and are flooded the rest of the year, compared to other populations. Three types of Bs (Figure 2) were observed, both in terms of the position of the centromeres and the arrangement of the heterochromatin. All three types were found only in population 8, while type I was observed in different variants and combinations in all populations in which Bs were present.

Sometimes, the species with Bs have a higher survival value than those without Bs [18,22], probably thanks to the genes present on these chromosomes. Therefore, the presence of active ribosomal genes on Bs has already been observed in plants [18,23,48,49,50,51], as is the case in this study (Appendix A).

The number and effects of B chromosomes in *N. poeticus* probably depend on the environmental conditions of the populations studied, as in the case of *Allium schoenoprasum* [30]. Our results may indicate a possible relationship between the presence of different types of B chromosomes in *N. poeticus* individuals and their exposure to water stress in the populations studied. Indeed, the population with three Bs types is characterized by the most stressful environmental conditions, especially by extreme combinations of water and temperature regimes in winter and summer.

### 3.4. Natural Triploids of Narcissus poeticus

Polyploidy, an important mechanism of adaptation and speciation in plants, is present in 47–70% of angiosperms [52]. Triploid progeny often survive and can act as ‘bridges’ in the establishment of higher ploidy levels [53,54]. Newly formed autopolyploids often possess novel physiological, ecological or phenological traits and can provide a rapid means of adaptation and speciation [52].

The genus *Narcissus* is characterized by sporadic polyploidization, with the highest frequency of tetraploids [5,8,13,16,55]. Polyploids may be an important factor in the origin of garden varieties, with triploids arising first, followed by tetraploids [8]. Triploids of unclear origin have been observed in several daffodil species, including *N. poeticus* cv. ‘Hellenicus’ and cv. ‘Pheasant’s Eye’ [8,13,16]. No wild populations are known for the triploid *N. poeticus* ‘Hellenicus’ and *N. poeticus* ‘Recurvus’, although they have been described in Greece and Switzerland. Wylie [8] suggested that these triploids are either outliers from cultivation or that they were selected by nature due to their triploid vigor.

The occurrence of natural triploids of *N. poeticus* in some Balkan populations was observed for the first time in this study. Morphologically, these individuals were indistinguishable from diploids. They possess three identical copies of a basic chromosome set, which speaks for their autotriploid origin. It is interesting to note that these triploids occurred in two almost completely contrasting habitats (populations 8 and 12) and also possessed 0–3 B chromosomes, with the types varying in karst population 8. Unfavorable environmental conditions influence both the occurrence and the developmental dynamics of polyploids. It is very difficult to determine which environmental factors have a stressful effect on plants, although numerous and rapid temperature fluctuations in humid environments are most commonly suggested [52,56]. Since the highest number of triploids was observed in population 8, it can be assumed that the production of diplo-pollen (frequently observed) in this population could be one of the consequences of specific and deleterious ecological conditions.

### 3.5. Deviant Forms of the Standard Karyotype

Spontaneous chromosomal translocations have already been observed in some species of the genus *Narcissus* [15,57]. In this study, the presence of individuals with chromosomal rearrangements (translocation or inversion) was observed in subalpine and serpentine populations where ecological conditions are the harshest (Appendix A, Table 1).

The observed inversion could be detected thanks to the presence of markers such as DAPI-positive bands and 5S rDNA loci on the inverted part of the chromosome. The other two deviant karyotype forms are most likely the result of a rearrangement of several chromosomes.

The individual from the serpentine population possessed an additional subterminal 35S rDNA locus on only one homolog of pair 3. The origin of this additional rDNA locus can be explained by spontaneous B-A translocation, which has already been described for some species [18,23,58,59]. This type of translocation is very rare, as only a small and particular piece of B-specific chromatin can cross the strong barrier between A and B chromosomes [59]. This specific event may be present in the individual from the serpentine substrate, especially due to two observed phenomena in this population: B chromosomes are characterized by the presence of active rRNA genes, and this individual had larger amounts of DNA than individuals without Bs.

### 3.6. DNA Content Increases Due to the Presence of B Chromosomes and Triploidy

This study found intraspecific variation in DNA amount between individuals with and without Bs. This result is consistent with the conclusions of Jones et al. [25] that Bs are a major source of intraspecific variation in amounts of nuclear DNA. The mean value obtained for diploid individuals without Bs (25.81 pg) is very similar to the values reported by Siljak-Yakovlev et al. [37] (24.33 pg) and Zonneveld [5] (26.0 pg). All three values obtained via flow cytometry are lower than the value obtained by Olszewska and Osiecka [38] (27.50 pg) using a different method, Feulgen densitometry. Comparing our data with those of Zonneveld [5], it can be assumed that his four diploid accessions with a genome size of more than 26.0 pg probably have at least one B chromosome. In addition, Zonneveld [5] found that Bs, in some species from the sections *Narcissus* and *Pseudonarcissus,* probably slightly contribute to the total DNA content.

The estimated mean genome size of the natural triploid Balkan individuals was 34.55 pg, which is about 4 pg compared to cultivated triploids measured by Zonneveld [5]. However, the genome size of the triploid individual with three Bs in this study is very close to the values determined by Zonneveld [5]. It might be interesting to compare the karyotypes and banding patterns of these Zonneveld accessions with our triploids to determine the cause of this difference in genome size.

## 4. Materials and Methods

### 4.1. Plant Material

The plant material (fresh leaves for genome size determination and bulbs for cytogenetic studies) of all 13 populations studied was collected in the field under different environmental and soil conditions. The localities, substrates, GPS coordinates (latitude and longitude), altitudes and numbers of individuals studied are listed in Table 3.

Voucher specimens are deposited in two herbaria: Herbarium of the National Museum of Bosnia and Herzegovina (SARA) and Herbarium of Mountain and Sea Institute in Makarska, Croatia (MAKAR). The living material was kept in the greenhouse of the University of Paris-Sud, France (Orsay Centre), until the actively growing root-tip meristems were obtained for chromosome preparation.

### 4.2. Genome Size Assessment

Total nuclear DNA was determined via flow cytometry following the technique of Marie and Brown [60]. *Triticum aestivum* ‘Chinese spring’ (2C = 30.90 pg) or *Artemisia arborescens* ‘Crete’ (2C = 11.43 pg) were used as internal standards. For nuclei extraction, daffodil and wheat leaves were chopped together with a razor blade in Galbraith buffer [61] containing 0.1% (*w*/*v*) Triton X-100 with addition of fresh 10 mM sodium metabisulfite, 1% polyvinylpyrolidone and RNAse (2.5 units/mL; Roche, Meylan, France). Nuclei were stained with DNA intercalating fluorescence dye propidium iodide (Sigma Aldrich Chimie S.a.r.l, Saint-Quentin-Fallavier, France) in final concentration of 70 µg/mL. The suspension was filtered through nylon mesh (pore size 48 µm) and kept at 4 °C. Measurement was made after at least 20 min of incubation. DNA content of at least 5000 stained nuclei was performed for each sample using a Partec CyFlow 532 nm laser cytometer (Beckman Coulter, Krefeld, Germany). The 2C DNA values of *Narcissus* specimens were calculated by assuming a linear relationship between propidium iodide fluorescence and nuclear DNA. At least five individuals per population were measured separately and with repetition.

### 4.3. Chromosome Preparation

For the cytogenetic analysis, root-tip meristems obtained from at least five bulbs per population were pretreated with 0.05% colchicine aqua solution for 3.30–4.30 h at 20 °C and fixed in ethanol/acetic acid (3:1) for at least 24 h at 4 °C [62]. Fixed root-tips were stored in the fixative at 4 °C until used.

### 4.4. Feulgen Staining and Karyotype Features

Root-tips were hydrolyzed in 1N HCl at 60 °C for 14 min, stained with Schiff reagent for at least 30 min and squashed in a drop of acetic carmine to check the number and morphology of chromosomes. For chromosome measurements and analysis of karyotypes, at least 10 well-spread chromosome plates per population were chosen.

Determinations of centromere position and chromosome type were performed following nomenclature of Levan et al. [63]. Total chromosome length (TCL), centromeric index (Ci), arm ratio (r), asymmetry index (AsI %, according to Arano and Saito [64]) and ratio of shortest/longest chromosome pair (R) were used for characterization of karyotypes. The length of the satellite was not included in measurements. In order to draw idiograms, the chromosome pairs are arranged by decreasing size and by arm ratio.

### 4.5. Silver Staining

Silver staining was used to detect the activity of 35S-5.8S-26S rDNA genes. Silver nitrate nucleoli staining was performed following the method of Hall and Parker [65]: young roots were put in 2% silver nitrate for one night at 60 °C. Under a light microscope, one hundred nuclei were checked for each population.

### 4.6. Slide Preparation for Fluorochrome Banding and FISH

Fixed root tips were washed in citrate buffer (pH 4.6) for 10 min and then digested in an enzymatic mixture (4% cellulase Onozuka RS (Yakult Pharmaceutical Industry Co., Tokyo, Japan), 1% pectolyase Y23 (Seishin Pharmaceutical Co., Tokyo, Japan), 4% hemicellulase (Sigma Aldrich Chimie S.a.r.l, Saint-Quentin-Fallavier, France)) in the same buffer at 37 °C for about 15 to 20 min (depending on root size). Meristems were spread on a clean slide in 45% acetic acid. For fluorochrome banding and FISH, slides with sufficient numbers of well-spread metaphase chromosome plates were selected. Slides were frozen at −80 °C during 24 h, and after cover-slip removal, they were air-dried for at least 24 h.

### 4.7. Fluorochrome Banding

For detections of GC-rich DNA regions, specific staining with chromomycin A3 (CMA; Sigma Aldrich Co., Steinheim, Germany) was performed following Schweiser [66] with minor modifications [67]. Slides were incubated for 10 min in McIlvaine’s pH 7 buffer with 5 mmol/L MgSO_4_, then stained with chromomycin A3 (0.2 mg/mL) for 90 min. The counter-stain methyl green (0.1%) was used during 7 min for the best contrast. After rinsing in the McIlvaine’s pH 5.5 buffer, the slides were mounted in glycerol anti-fade solution (Citifluor AF2, Agar Scientific, Stansted, UK).

Identification of AT rich regions was performed with Hoechst 33258 (2.2-(4-hydroxyphenyl)–6–benzimidazolyl–6-(1-methyl4-piperazyl)–benzimidazol-trihydrochloride; Ho; Sigma-Aldrich Co., Steinheim, Germany) staining according to Martin and Hesemann [68], with minor modifications. Slides were rehydrated for 5 min in each of the graded ethanol series (70%, 50%, 30%), washed in distilled water and incubated in McIlvaine’s pH 5.5 buffer for 10 min. After a 2 min long staining with Hoechst (2 μg/mL), slides were rinsed for 15 min both in McIlvaine’s pH 5.5 buffer and distilled water, and mounted in glycerol anti-fade solution (Citifluor AF2, Agar Scientific, Stansted, UK).

The slides with well-stained chromosome plates were destained in 3:1 ethanol/acetic acid, dehydrated in a graded ethanol series (70%, 90%, 100%) and used for other types of staining.

### 4.8. Fluorescence In Situ Hybridization (FISH)

A double FISH experiment was carried out with two DNA probes following the technique of Heslop-Harrison et al. [69] with minor modifications [66]. The 35S rDNA probe was a clone of a 4 kb EcoRI fragment containing a part of 35S, 5.8S and 26S rDNA from *Arabidopsis thaliana*, labeled with direct Cy3 fluorochrome (Amersham Co., Courtaboeuf, France), and pTa 794 probe [70] was a clone containing a 410-pb BamH1 fragment of 5S rDNA from wheat, labeled with digoxigenin-11-dUTP (Roche Diagnostics, Meylan, France). Chromosomal DNA and probes were denatured at 72 °C for 10 min and then put at 55 °C for 5 min. Hybridization was conducted at 37 °C overnight. Slides were washed in stringent wash 2xSSC, 0.1xSSC and 4xSSCT, revealed by anti-digoxigenin-11-dUTP, counter-stained and mounted in Vectashield mounting medium with DAPI (Vector Laboratories, Peterborough, UK).

Observation of the chromosome plate was performed with an epifluorescence Zeiss Axiophot microscope with various combinations of Zeiss excitation and emission filter sets (01, 07, 15 and the triple filter set 25). Hybridization signals were analyzed using a highly sensitive CCD camera (Retiga 2000R, Princeton Instruments, Evry, France) and an image analyzer (Metavue, vers.7.0r4, Evry, France). The images were processed in Adobe Photoshop CS4 (Adobe Systems, Zurich, Switzerland) for color contrast and brightness.

### 4.9. Data Analysis

Data were analyzed using Statistica 7 for Windows using a one-way analysis of variance (ANOVA). After analysis of variance, in order to determine significant inter- and intrapopulation differences Duncan’s post hoc test was used.

## 5. Conclusions

In our panel of 13 natural populations of *Narcissus poeticus* from the Balkan region, several important observations can be emphasized. Despite the geographical distances between the populations and the different types of habitats in which they grow, the standard characteristics of karyotype and heterochromatin organization were constant at the species level. Variation in 2C DNA values due to the presence of polyploidy (some triploid individuals were observed in two populations) and B chromosomes (in 9 out of 13 populations) was detected. The presence of B chromosome polymorphism and the activity of the rRNA genes they carry were evident. In addition, individuals with B chromosomes appear to have some selective advantages. Considering the population size and the number of individuals of poet’s daffodil in the studied populations, it can be assumed that the most optimal living conditions for this species, in the region where the present study was conducted, are in the karst fields zone in the continental Dinaric Alps.

## Figures and Tables

**Figure 1 plants-13-00208-f001:**
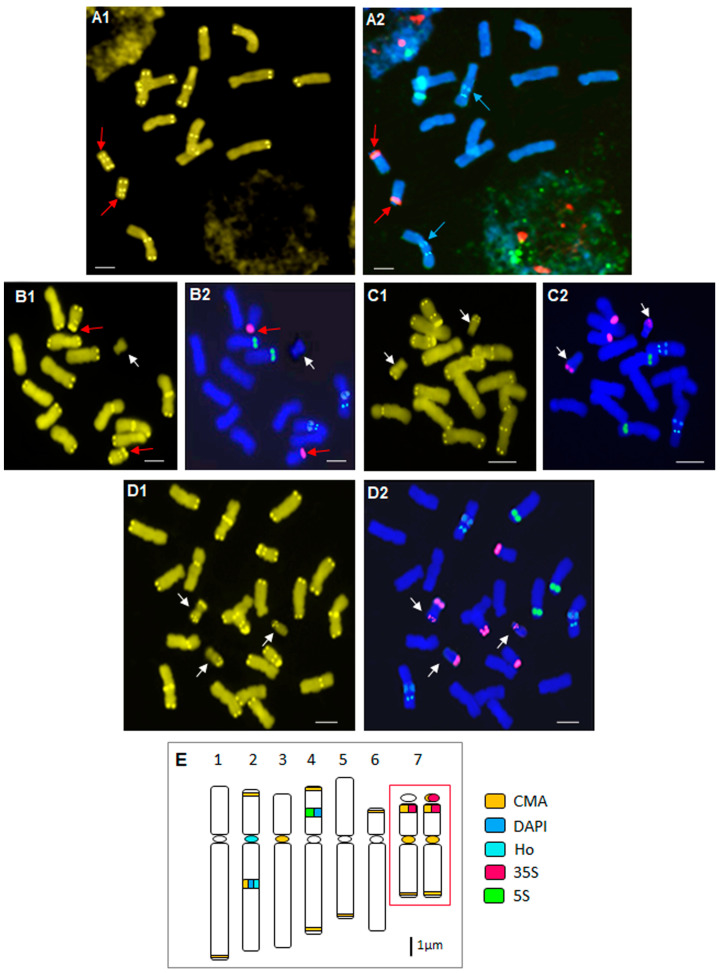
Results of chromomycin banding and FISH of the standard karyotype for different populations of *N. poeticus*. (**A1**) CMA and (**A2**) FISH for diploid individuals with standard karyotype without Bs. Chromosome pair 7 shows very intense CMA and 35S rDNA signals only at the secondary constriction on the short arm, with no staining on the satellite (red arrows). The blue arrows indicate the Hoechst staining colocating with the CMA band. (**B1**) CMA and (**B2**) FISH for diploid individuals with standard karyotype and 1 B chromosome (white arrows; populations 5–10). Chromosome pair 7 showed very intense CMA and 35S rDNA signals on both sides of the secondary constriction (red arrows). (**C1**) CMA and (**C2**) FISH for diploid individuals with 2 Bs (white arrows; population 11). (**D1**) CMA and (**D2**) FISH for triploid individuals with 3 different types of Bs (white arrows, serpentine population 8). (**E**) Idiogram corresponding to the standard diploid karyotype (two homologous chromosomes of pair 7 show the difference in satellite staining methods (boxed)). Scale bar on photographs = 5 µm.

**Figure 2 plants-13-00208-f002:**
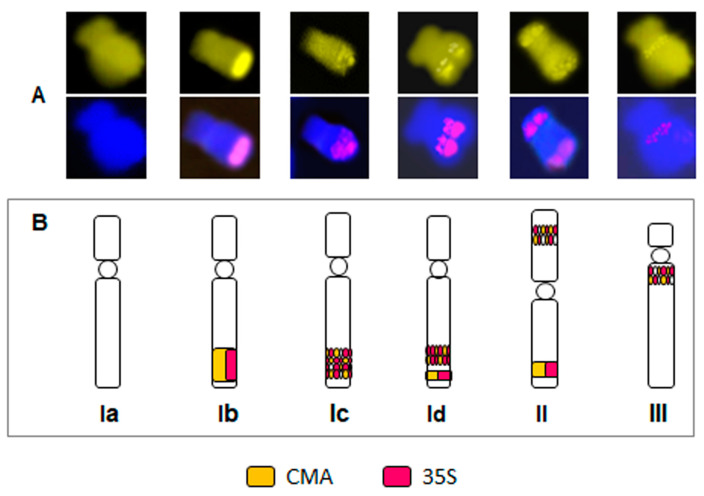
Bs polymorphism. (**A**) CMA and FISH images of different types of Bs chromosomes. (**B**) Idiograms corresponding to different types of B chromosomes: (Ia) submetacentric Bs with no signals, (Ib–Id) submetacentric Bs with different signal pattern (colocalization of CMA and 35S rDNA in block Ib, dispersed Ic, or with both patterns Id); (II) metacentric Bs with two signals on both arms, one in block and the other dispersed; and type (III) subtelocentric Bs with a dispersed signal on the long arm near the centromere.

**Table 1 plants-13-00208-t001:** Summary of the results: chromosome number (2n) and presence of B chromosomes and chromosomal translocations; characteristics of chromosome pair 7; type of Bs; number of nucleoli after AgNO_3_ staining; ecological characteristics of the populations analyzed.

Acc. No	2n	Pair 7		Bs Type	Number of Nucleoli	Ecological Characteristics
		CMA	35S			
1.	14	+/−	+/−	/	1–2	A
2.	14 ^T^	+/−	+/−	/	1–2	
3.	14	+/−	+/−	/	1–2	
4.	14			/	1–2	
5.	14 + 0 − 1B	+	+	Ib, Ic	1–3	B
6.	14 + 0 − 1B	+	+	Ib, Ic	1–3	
7.	14 + 0 − 1B	+	+	Ia, Ib, Ic	1–3	
8.	14 + 0 − 3B 21 + 0 − 3B	+	+	Ib, Ic, Id, II, III	1–6	
9.	14 + 0 − 1B	+	+	Ic, Id	1–3	C
10.	14 + 0 − 1B	+	+	Ia, Id	1–3	
11.	14 ^T^ + 0 − 2Bs	+	+	Ib, Ic, Id	1–4	D
12.	14 ^T^ + 0 − 1B 21	+	+	Ib, Ic, Id	1–3	
13.	14 + 0 − 1B	+	+	Ib, Ic, Id	1–3	

CMA, chromomycin bands; 35S, 35S rDNA loci; Bs type, see Figure 1; ^T^, chromosomal translocations; ecological characteristics: (A) submountain and subalpine wet meadows in continental Dinaric Alps, short vegetation season, long-term snow cover, pronounced lower temperatures, late frosts in spring and early autumn; (B) moderate-continental and mountainous climate, karst fields, very dry in summer and flooded the rest of the year, long-term snow cover; (C) littoral Dinaric Alps, Mediterranean climate, water stress, high summer temperature and insolation, strong winds; (D) North-eastern Belt of Dinaric Alps, moderate-continental climate, fairly cold winters and hot and dry summers, heavy metals, high insolation.

**Table 2 plants-13-00208-t002:** Genome size of the different karyotype variants of *N. poeticus* and Bs contribution to the total 2C DNA values.

	Diploids					Triploids	
				Translocation			
	Without Bs (Pops 1–4)	With 1B (Pops 5–7,9,10,12,13)	With 2Bs (Pop 11)	Serpentine (Pop 12)	Limestone (Pop 2)	Without Bs (Pops 8,12)	With 3 Bs (Pop 8)
2C DNA (pg ± sd)	25.25 ± 0.39	25.81 ± 0.36	26.52 ± 0.15	25.77	24.91	34.55 ± 0.59	38.80
Bs contribution to total 2C DNA (%; ±sd)		2.22 ± 0.27	5.03 ± 0.09				12.30

For diploids, mean values are calculated on the basis of 39 individuals without Bs, 27 individuals with 1 B, and two individuals with 2 Bs, as well as three triploid individuals without Bs. In all other cases, individual values are given; sd, standard deviation; % for Bs, percentage of genome size concerning B chromosomes.

**Table 3 plants-13-00208-t003:** Geographical origin of the analyzed populations of *Narcissus poeticus*.

Acc. No	Locality	Substrate	Latitude (N)	Longitude (E)	Altitude (m)	N
1.	Mt. Čvrsnica, Mala Čvrsnica, B&H	L	43°34′38″	17°29′33″	1384	5
2.	Mt. Čvrsnica, Barice, B&H	L	43°35′01″	17°30′02″	1350	6
3.	Glamočko Polje, Mliništa, B&H	L	44°14′07″	16°49′57″	1198	11
4.	Mt. Zelengora, Točila, B&H	L	43°19′51″	18°33′10″	1470–1570	11
5.	Livanjsko Polje, Peulje, B&H	L	44°08′42″	16°27′56″	806–859	13
6.	Livanjsko Polje, Čelebići, B&H	L	43°56′01″	16°40′52″	710	5
7.	Mt. Čvrsnica, Rakitno, B&H	L	43°32′79″	17°24′90″	895–910	8
8.	Gatačko Polje, Avtovica, B&H	L	43°09′15″	18°31′32″	940	54
9.	Mt. Biokovo, Lađena, Cro	L	43°17′27″	17°05′25″	1265	7
10.	Mt. Biokovo, Vošac, Cro	L	43°18′46″	17°04′01″	1340	9
11.	Kladanj, Katranica, B&H	S	44°16′09″	18°33′15″	800–940	26
12.	Žepče, Matinski Vis, B&H	S	44°28′03″	17°58′37″	820–900	23
13.	Mt. Zlatibor, Mokra Gora, Srb	S	43°49′09″	19°30′24″	810	11

B&H, Bosnia and Herzegovina; Cro, Croatia; Srb, Serbia; L, limestone; S, serpentine; N, number of individuals analyzed.

## Data Availability

Data are contained within the article.

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
