# Peer review of "Karyotype Variability in Wild Narcissus poeticus L. Populations from Different Environmental Conditions in the Dinaric Alpsâ€"

_plants, 2024, doi:10.3390/plants13020208_

Round 1
Reviewer 1 Report
Comments and Suggestions for Authors
1. You say that the species is very variable morhologiclly. How variable were the plants from the 13 populations that you collected?
2. Chromosome length means very little. The chromosome get shorter when the cells are treated wityh colchicine.
3. I would expect various genome sizes in leaf cells. 2c values in the lowest peak but also repliated and replicating nuclei. Also I would expect some endoployploid nuclei.
4. Explain the reasons for you suggestion of selective advantage for the B chromosomes.
Comments on the Quality of English Language
1. Samuel Pyke needs to d a lot more work on the English language usage.
Reviewer 2 Report
Comments and Suggestions for Authors
The manuscript described fluorochrome banding and FISH with rDNA sequences to 13 different populations of Narcissus. Using these methods, both numerical and structural chromosome mutations were revealed in the analysed taxon. Additionally, three different types of B chromosomes were observed. Although the good quality of the data presented, especially the microscopic images, the manuscript requires thorough revision. The description of the results should be improved (see my comments below), and the discussion shortened and reworded. This part of the manuscript mainly contains various information about B chromosomes and karyotype evolution, but provides little discussion of the presented results. There are also many spelling and grammar mistakes which needs to be corrected.
Page 2 lines 87-89
“In all investigated populations N. poeticus presented the diploid chromosome number 2n=2x=14, except in two populations (8 and 12; Tables 1 and 4) where there also exist triploid individuals, 2n=3x=21.”
There is no chromosome number in any of these tables. This information should be presented in the description of the results. While the data from morphometric analyses of chromosomes could be transferred to supplementary data. The typical idiogram of diploid and triploid populations could be presented.
Page 2 line 93
“2n=8 sm + 2 m + 2 st + 2 sm-msat”
Does it mean that there was polymorphism in chromosome morphology? If Yes, please give the most common one and then describe the variations.
Page 3; Lines 104-105
Please add the images of all observed variants to the supplementary data.
Figure 1
1D – add the images of the karyotype, which showed no 35S rDNA signals in the chromosome fragment distal to secondary constriction (satellite) .
The figure is not described properly.
- There are three different types of signals in A2, C2, B2, F2, G2 and H2. Explained which color corresponds to which probe- The whole chromosomes are counterstained with DAPI. The positive DAPI bands are not shown. What does it mean the navy blue band on idiograms?
- 1E – why do some chromosomes have yellow/red bands while others show many small yellow/red balls?
Page 5, line 114 – I have to admit that I can see the disappears 35S rDNA signals in this Figure
Page 5, line e.g. 109 “for diploid serpentine individuals”e.t.c.
Please describe the accessions as in Table 4 (Materials Methods).
Table 2 – the name/labelling of the analysed population, according to Table 4 (MM section) should be used
Page 6, lines 154-157
This fragment should be transferred to Materials and Methods
Page 6, line 173
To detect the constitutive heterochromatin the C-banding should be used. CMA3 often stained locus of 35S rDNA – even if it is transcriptionally active.
Page 7, lines 192-195
Show the results of silver staining. Have you observed only a number of nucleoli or the Ag-NORs on B-chromosomes were also observed?
Page 7 line 206-207
” This inversion could be detected thanks to the presence of markers such as DAPI and 5S rDNA signals on the inverted part of the chromosome”
It is possible to write about DAPI positive/negative bands but not about signals.
Page lines 210-219
In the case of an individual presented in fig 1F, the results suggesting translocation are presented, while in cases of individuals from 1G and especially 1H the translocation can only be hypothesised. It would be better to transfer this discussion to the “Discussion” section.
Discussion
Lines 265-344
This fragment looks more like an introduction or part of the review article. It is not the proper discussion. This fragment should be shorter, and the diversity of B chromosomes should be discussed. Is it possible to propose any hypothesis of their origin?
Lines345-372
Does Narcissua propagate not only generative but also in a vegetative way? Does vegetative reproduction influence the persistence of triploid or forms in natural populations?
The same question concerns the cytotypes with chromosome rearrangements.
Lines 378-385
The number and localisation of rDNA loci can be very variable, even among individuals of one population. Many reports showed this phenomenon in many different species both monocots and eudicots.
Lines 386-411
It is obvious that additive chromosomes cause an increase in genome size. This part should be shortened.
Olszewska and Osiecka (1982) used different methods (densitometry) to estimate C-value, while flow cytometry was used in the present manuscript.
Materials and Methods
Page 12, lines 439-443
There is only a description of material pretreatment and fixation. Please, explain how the slides were prepared.
Round 2
Reviewer 1 Report
Comments and Suggestions for Authors
Needs a final revision of the English language usage.
Comments on the Quality of English LanguageNeeds a final revision of the English language usage.
Author Response
Needs a final revision of the English language usage.
Response: Done.
Reviewer 2 Report
Comments and Suggestions for Authors
Authors made most of the suggested change. I just have a few comments:
Line 93-94
“although this is not the case in the Anthoxanthum aristatum/ ovatum complex [33].”
This information is not connected with the manuscript.
Table 1 is placed on page 6, but it is mentioned for the first time on page 3,
Figure 1 – this is 35S rDNA locus or 5S rDNA locus – not “35S” or “5S” – please correct the whole manuscript
Page 7, lines 192-195
Show the results of silver staining. Have you observed only a number of nucleoli or the Ag-NORs on B-chromosomes were also observed?
“Response: We did not perform Ag staining on the chromosomes. However, FISH shows the presence of rRNA genes on Bs. As we found more nucleoli in individuals with Bs bearing 35 rDNA, we assume that rRNA genes on Bs were active.
The number of formed nuclei increased and corresponded to the total number of 35S rDNA loci found on A and B chromosomes in the case of individuals with Bs carrying rRNA genes(See Figure S2).”
The nucleoli number is very indirect evidence of the number of active 35S rDNA loci. It would be much better to show silver staining on chromosomes or show that in the interphase nucleus, there are more than two decondensed FISH signals which are connected with nucleolus/nucleoli to prove the transcriptional activity of 35S rDNA locus on B chromosomes.
Regarding the “discussion”, I will also add that the maximum number of nucleoli can be related to the number of active 35S rDNA loci. The nucleoli tend to fuse. And they can also be fragmented, e.g. during ageing and PCD
Author Response
Authors made most of the suggested change. I just have a few comments:
Line 93-94
“although this is not the case in the Anthoxanthum aristatum/ ovatum complex [33].”
This information is not connected with the manuscript.
Response: This part of the sentence has been deleted.
Table 1 is placed on page 6, but it is mentioned for the first time on page 3,
Response: The table 1 has been moved to the page 3.
Figure 1 – this is 35S rDNA locus or 5S rDNA locus – not “35S” or “5S” – please correct the whole manuscript
Response: Thank you. Done.
Page 7, lines 192-195
Show the results of silver staining. Have you observed only a number of nucleoli or the Ag-NORs on B-chromosomes were also observed?
Response 1 revision: "We did not perform Ag staining on chromosomes. However, FISH showed the presence of rRNA genes on Bs. As we found more nucleoli in individuals with Bs bearing 35 rDNA, we assume that rRNA genes on Bs were active.
The number of formed nuclei increased and corresponded to the total number of 35S rDNA loci found on A and B chromosomes in the case of individuals with Bs carrying rRNA genes(See Figure S2).”
The nucleoli number is very indirect evidence of the number of active 35S rDNA loci. It would be much better to show silver staining on chromosomes or show that in the interphase nucleus, there are more than two decondensed FISH signals which are connected with nucleolus/nucleoli to prove the transcriptional activity of 35S rDNA locus on B chromosomes.
Regarding the “discussion”, I will also add that the maximum number of nucleoli can be related to the number of active 35S rDNA loci. The nucleoli tend to fuse. And they can also be fragmented, e.g. during ageing and PCD
Response: We added Figure S1B of representative interphase nucleus with three active 35S rRNA loci. Two rDNA loci on the chromosome A show high activity, while the locus on chromosome B shows the beginning of activity. We have observed other INs with rDNA activity on the B chromosome but unfortunately we do not have any other photographs except the one shown in figure S1.